# Radar versus optical: The impact of cloud cover when mapping seasonal surface water for health applications in monsoon-affected India

**Gowri Uday**[1☯], **Bethan V. Purse**[2‡], **Douglas I. Kelley**[2‡], **Abi Vanak**[1,3], **Abhishek Samrat**[1,4‡], **Anusha Chaudhary**[1,5], **Mujeeb Rahman** [ID][1], **France F. Gerard** [ID][2☯]*

**1** Ashoka Trust for Research in Ecology and the Environment, Bengaluru, India, **2** UK Centre for Ecology and Hydrology, Crowmarsh Gifford, Wallingford, United Kingdom, **3** School of Biological Sciences, University of KwaZulu-Natal, Durban, South Africa, **4** School of Engineering and Computing, University of Central Lancashire, Preston, United Kingdom, **5** Department of Geography, University of Florida, Gainesville, Florida, United States of America

☯ These authors contributed equally to this work.
‡ BVP, DIK and AS also contributed equally to the work.
* ffg@ceh.ac.uk

**Data Availability Statement:** Mapped water body files are available from EIDC: Uday, N.G.; Purse, B.;

## Abstract

Surface water plays a vital role in the spread of infectious diseases. Information on the spatial and temporal dynamics of surface water availability is thus critical to understanding, monitoring and forecasting disease outbreaks. Before the launch of Sentinel-1 Synthetic Aperture Radar (SAR) missions, surface water availability has been captured at various spatial scales through approaches based on optical remote sensing data. A critical drawback of the latter is data loss due to cloud cover, however few studies have quantified this. This study evaluated data loss due to clouds in three Western Ghats (India) districts. These forest-agricultural mosaic landscapes, where water-related diseases are prevalent, experience the Indian monsoon. We compared surface water areas mapped by thresholding 10m Sentinel-1A SAR data with the optical 30m Landsat-derived Joint Research Centre (JRC) Global Surface Water product, currently the only globally available long-term monthly surface water data product. Backscatter thresholds were identified manually, and our Bayesian algorithm found these thresholds were very likely (>97%). While the Sentinel-1 SAR-based and JRC's optical-based approach mapped surface water extent with high overall accuracy (> 98%) when the cloud cover was low, the unmapped surface water area was substantial in the JRC product during the monsoon months. Across the districts, the average cloud cover in the July-August period was 92% or 90% for 2017 and 2018 respectively, resulting in 25% or 23% of the surface water area being unmapped. Also, the more detailed 10m resolution of Sentinel-1A SAR helped detect the many small water features missed by 30m JRC. Thus, for predicting water-related disease risks linked to small water features or monsoon rainfall, Sentinel-1A SAR is more effective. Finally, automatic backscatter thresholding for unvegetated surface water mapping can be effective if threshold values are adapted to regional-specific backscatter spatial and temporal variations.

Vanak, A.T.; Samrat, A.; Chaudhary, A.; Gerard, F. (2022). Radar derived seasonal surface water maps for three Indian districts (Shivamogga, Sindhudurg, Wayanad), 2017-2018. NERC EDS Environmental Information Data Centre. https://doi.org/10.5285/3c23fea1-5b27-4b01-b9ef-fc13346cfedc Field collected reference data are available from EIDC as part of https://catalogue.ceh.ac.uk/documents/cacb66de-aea0-41d5-97b3-9eacd4683aaf.

**Funding:** The MonkeyFeverRisk project that led to these results is supported by the Global Challenges Research Fund and funded by the MRC https://www.ukri.org/councils/mrc/, AHRC https://www.ukri.org/councils/ahrc/, BBSRC https://www.ukri.org/councils/bbsrc/, ESRC https://www.ukri.org/councils/esrc/ and NERC https://www.ukri.org/councils/nerc/ [grant numbers MR/ P024335/1 and MR/P024335/2], awarded to BVP, AV and FG. Additional support was provided from the IndiaZooRisk Project, which is funded by UK Research and Innovation https://www.ukri.org/councils/nerc/ through the Global Challenges Research Fund [MR/T029846/1] and by the NERC SUNRISE project [grant number NE/R000131/1]. DIK was supported by the Natural Environment Research Council as part of the NC-International programme [NE/X006247/1] delivering National Capability.

**Competing interests:** The authors have declared that no competing interests exist.

## Introduction

Surface water availability is a key driver of the transmission and spillover of diverse infectious diseases [1–4], including water-borne pathogens, water-based parasites, and vector-borne diseases [5–8]. Their burdens pose significant global challenges to Public Health [9, 10]. Thus, information on surface water availability is critical to improve our understanding of water-related disease burdens, provide operational tools to predict disease risks, and inform Public Health interventions [11, 12]. This has led to substantial interest in capturing surface water extent and dynamics from satellite remote sensing. Since surface waterbodies are highly variable over space (households to regions) and time (within and between years), so too are the functional resources they provide to pathogens, vectors, hosts, and people [13], and the resulting ecological interactions that underpin disease transmission and spill over to people [14, 15]. These ecological processes consequently shape the requirements for public health-related surface water mapping. For example, given the short (< 1 month) generation times and dispersal ranges of mosquito vectors, surface-water mapping must capture the highly ephemeral and seasonal nature interventions [14]. Capturing the impacts of extreme precipitation and drought events on surface water is also critical for understanding the dynamics of water availability to agents, hosts, and vectors of water-related diseases, e.g., [16–18]. Diverse types and sizes of waterbodies may provide functional resources to species within transmission cycles of water-related diseases and thus need to be mapped to predict disease patterns. For example, in Senegal, where key mosquito vectors of Rift Valley Fever use small, vegetated ponds, [12] captured the spatial-temporal evolution of these ponds using the Normalised Difference Pond Index and the Normalised Difference Turbidity Index derived from optical SPOT-5 images [19]. Moreover, as highlighted for Schistosomiasis [7], when predicting disease patterns from remotely sensed data, it is also essential to try to bridge the spatial gap between the habitats where transmission and spillover occur (here for the parasite and intermediate host) and the community location where human surveillance and intervention takes place. One means of achieving this is using high-resolution data scaled to the habitat use of the people, animals, and vectors involved in the transmission cycle [13, 20].

Against this backdrop of requirements, some studies have correlated terrain indices (e.g., topographic wetness index and topographic position index) with small waterbodies [21], and most have developed optical remote sensing approaches to relate surface water and wetness to patterns in water-related infectious diseases [7, 14, 19, 22]. Though open waterbodies strongly absorb radiation in the Near Infrared (NIR) band [23], to increase mapping success or allow for turbidity and aquatic vegetation, optical-based surface water mapping tends to use indices combining different spectral bands [12, 19, 22, 24]. For example, the Normalised Difference turbidity index (NDTI) combines a red and green band [12], and the Normalised Difference Water index (NDWI) or Normalised Difference Pond Index (NDPI) combines a green and NIR band [12, 24]. The related Modified Normalised Difference Water Index (MNDWI) uses a green and Short-Wave Infrared (SWIR) band (instead of NIR) to improve the water mapping in landscapes dominated by built-up areas. Some indices have been used at a low resolution (~500m to 1km), derived from long-term, multi-temporal MODIS or AVHRR data, to link continental or national patterns in wide-ranging water-related diseases to seasonal surface water and wetness. For example, MNDWI has been used to predict mosquito-borne West Nile Virus patterns across Europe [25, 26]. At local scales, when derived from high-resolution optical imagery such as SPOT and LANDSAT, these indices have been used to accurately predict the location of many different types of surface waterbodies and related risks from water-borne, water-based [7, 27] and vector-borne diseases [12, 22], across wide-ranging of geographical contexts. However, while high-resolution optical datasets can detect the location of relevant

small waterbodies more accurately [28], data is only acquired a few times a year [12]. Some disease-related applications in arid zones have addressed the need for high-frequency season data to inform interventions by integrating infrequent high spatial resolution EO data with frequent medium-resolution datasets (e.g., MODIS) to monitor the dynamics of waterbodies [19].

However, a key drawback of optical data is that cloud cover affects optical sensors, and data loss through cloud cover can be substantial in sub-tropical and tropical zones with pronounced wet and dry seasons [29]. In such zones, optical satellite imagery may be unable to detect waterbodies exhaustively [30] or detect epidemiologically relevant changes in extent between the wet and dry seasons. Radar imagery, by contrast, is not impacted by cloud cover and atmospheric conditions, and since 2017, has been available globally at an enhanced spatial resolution (~10m) and temporal frequency (6–12 days revisit period) from the Copernicus Sentinel-1 satellites. The Sentinel-1 synthetic aperture radar (SAR) could potentially capture epidemiologically relevant dynamics of many waterbodies. Open waterbodies, in particular, are distinguishable from other land cover types due to their typical low backscatter response, and Sentinel-1A SAR has already been used to characterise surface water extent [31] and surface water dynamics, including waterbodies utilised by mosquitoes [14, 32], and to map flooding [33–35], and rice paddies [36].

This study used 10m Sentinel-1 C-band SAR data to produce high-resolution waterbody maps in monsoon-affected regions of rural India where water-related diseases, such as Leptospirosis and Japanese Encephalitis, impact human communities. Specifically, we wanted to establish if SAR data provides seasonal maps of disease-relevant small waterbody habitats ($< 900m^2$). Also, to quantify the impact of cloud cover on an optically derived alternative map of surface waterbodies, we compared our mapping results to the European Commission Joint Research Centre's global surface water product. The European Commission Joint Research Centre's global surface water product is derived from Landsat data and has a similar spatial resolution (i.e., 30m). It is currently the only global product that provides long-term (from 1984 onwards) monthly open water coverage [37].

## Methods

Fig 1 summarises the approach to creating, evaluating and comparing radar-based waterbody maps with the optical-based JRC global surface water product. First, we explored existing radar-based thresholding methods for mapping waterbodies (Fig 1A). We then compared the classification performance (Fig 1B) and derived waterbody metrics (Fig 1C) of the JRC global surface water product and the chosen manual thresholding method.

### Study area

To contrast between different seasonal surface water dynamics and land cover/use landscape patterns, we focussed our work on three districts located in the Western Ghats (India) (Fig 2), where a range of water-related diseases are prevalent (Table 1). Table 1 describes the area, topography, climate and water-related diseases found in the districts. Shivamogga (in Karnataka state) has a tropical climate and is the largest of the districts. It has a dense network of tributaries that feed six major rivers and includes areas with a high density of paddy fields. The western part of Shivamogga lies in the Western Ghat mountains and experiences heavy rainfall, while towards the east, the landscape is generally flat with low hills. Sindhudurg (in Maharashtra state) has a moist and humid climate. It has a highly uneven terrain with very narrow riverine plains that fringe the coastline. Five major rivers flow from the Western ghats in the East to Sindhudurg's coastal plain in the West. Wayanad (in Kerala state) is the smallest of the three districts and has a tropical monsoon climate. It is part of the mountainous plateau of the Western Ghats. In the east, the terrain is flat and open. Towards the centre are low-lying hills,

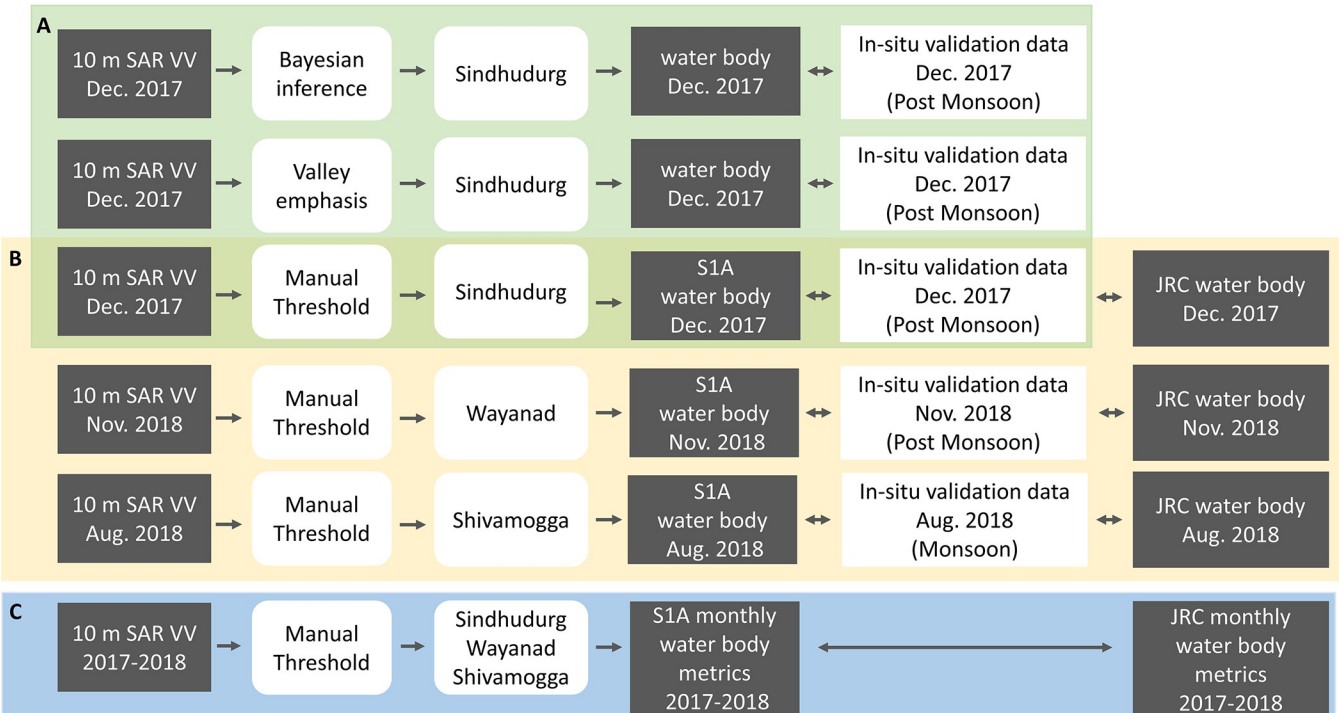

**Fig 1. Schematic summarising steps to evaluate and compare radar-based waterbody maps with the optical-based JRC global surface water product.**

while in the north are high mountainous hills. Paddy fields dominate the valleys and flat open terrain. Wayanad is rich in water resources, with a major east-flowing river, Kabini and Phookat Lake, which is perennial. Tributaries of the Kabini River are the Thirunelli River, Panamaram River and Mananthavady River.

## Data sources

We focussed on the years 2017 and 2018. For the waterbody mapping, we used Sentinel-1A SAR data captured in interferometric wide swath mode (IWS) with a ~10m resolution (from the Alaska satellite facility download page: https://search.asf.alaska.edu). We used the Level 1 Ground Range Detected (GRD) product in VV polarisation and projected it to ground range using an Earth ellipsoidal model. VV polarisation has a higher backscatter than other polarisations, and its higher sensitivity to surface roughness helps identify submerged agricultural fields and shallow waterbodies [43].

The JRC monthly water history is part of the JRC Global Surface Water product data bundle [37] and is available from Google Earth Engine (https://global-surface-water.appspot.com). It provides 30m monthly global maps of surface water from 1984 to date, updated annually and generated from optical Landsat data (i.e., Landsat5-TM, Landsat7-ETM and Landsat8-OLI). The monthly map contains 3 classes: 'No Data' (i.e., cloud), 'Not water' and 'Water'. We used monthly surface water maps for 2017, and 2018 from the GSW1_1 version (i.e., v1.1. or 1984–2018 versions) and will refer to the 2017–18 data bundle as 'JRC' hereafter.

## Pre-processing of Sentinel-1A SAR data

We pre-processed Sentinel-1A SAR data using the Sentinel application platform (SNAP). It involved a series of standard corrections, including applying an orbit file, thermal noise

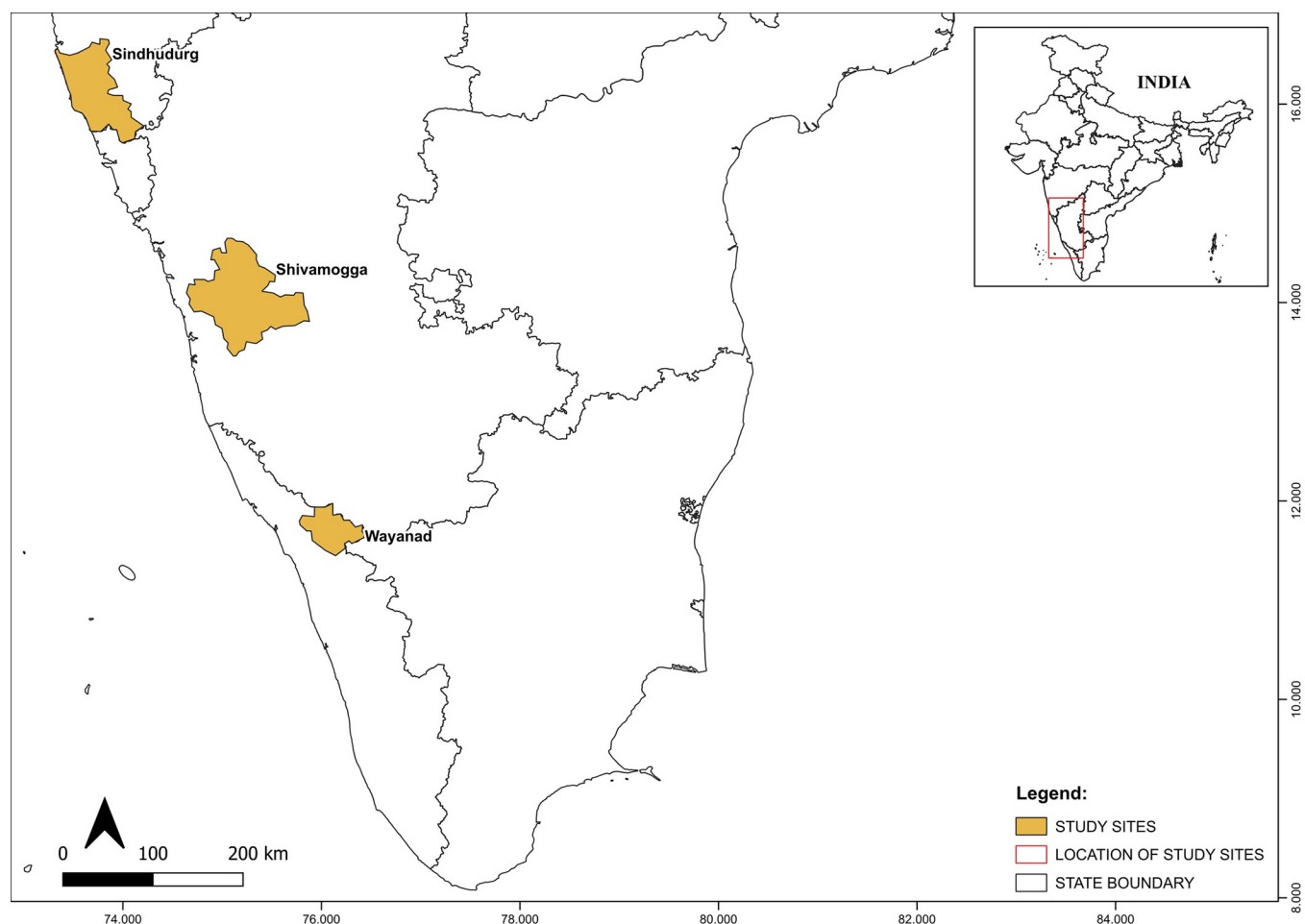

**Fig 2. Map showing the location of the three focal districts (Sindhudurg, Shivamogga and Wayanad), India.** Source of Administrative boundaries: https://github.com/HindustanTimesLabs/shapefiles/tree/master/india/district.

removal, calibration, and a speckle filter. Applying an orbit file updates the scene orbit state vectors, providing accurate satellite position and velocity information. Thermal noise removal helps to remove the additive noise [44] in sub-swaths, reducing discontinuities between sub-swaths for scenes in multi-swath acquisition mode. Calibration computes the backscatter intensity using sensor calibration parameters, and the subsequent speckle filtering removes backscatter image noise. Here, we used the refined Lee speckle filter applied to a 3 x 3 window

**Table 1. Description of the three focal districts.**

| District | Area, (km²) | Elevation (m) | Rainfall (mm) [Temperature (˚C)] | Major crop | History of water-related diseases |
|---|---|---|---|---|---|
| Shivamogga | 8,495 | 500–1340 | 1813 [21–28] | paddy, cotton, ragi, jowar, maise, sugarcane, pulses, sunflower and small-scale vegetables [38] | Kyasanur Forest Disease (KFD), Handigodu Syndrome (http://www.deccanherald.com/Content/May192007/state200705182509.asp), cholerae [39], Dengue [40], Japanese Encephalitis [40], |
| Sindhudurg | 5,207 | 0–700 | 3287 [17–35] | paddy, ragi, cashew nut, mango, and kokum [41] | KFD, Leptospirosis [40], Viral Hepatitis [40] |
| Wayanad | 2,132 | 700–2100 | 2322 [18–29] | paddy, coffee, tea, spices [42] | KFD, Viral Hepatitis—A [40], Cholera [40], Leptospirosis [40], Acute Diarrhoeal Disease [40], Rubella [40] |

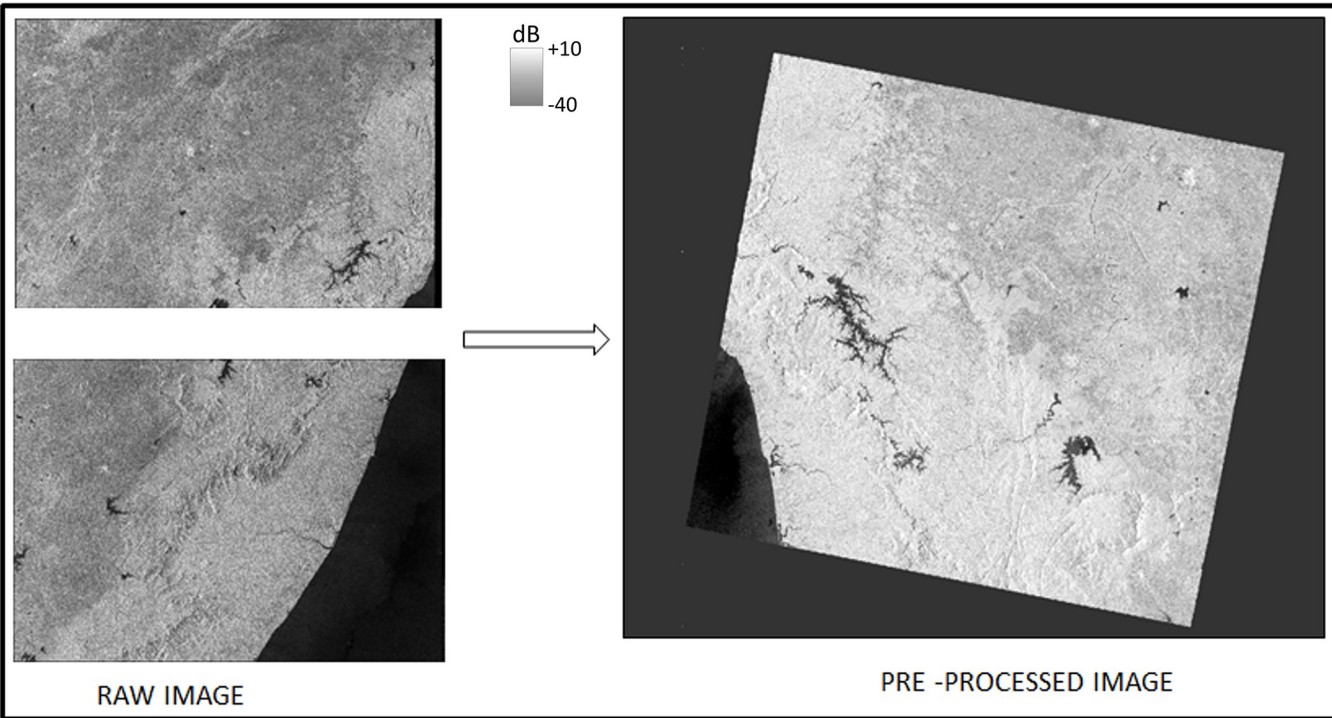

**Fig 3. Raw and pre-processed Sentinel-1A SAR VV backscatter image of Shivamogga of January 2018**

[45]. Each district stretched across two adjacent SAR scenes, so we mosaicked the adjacent scenes using the SNAP slice assembly tool and then cropped the resulting image to fit the corresponding district extent using the SNAP subset tool. For example, Fig 3 shows the raw and pre-processed imagery for Shivamogga January 2018.

Source: Copernicus Sentinel-1 data 2018, accessible via https://dataspace.copernicus.eu/explore-data/data-collections/sentinel-data/sentinel-1 or through GEE using ee.ImageCollection("COPERNICUS/S1_GRD")).

### Extraction of waterbodies

**Rationale.** Generally, we expect SAR backscatter from a waterbody to be significantly lower than the surrounding landscape due to the waterbody's low surface roughness and specular reflection [33]. Note that [46] found higher backscatter for waterbodies for low ($< \sim 30°$) radar incidence angles. Therefore, although supervised image classification could distinguish waterbodies from their surroundings, e.g., [31], a more straightforward and more effective approach is to identify a threshold backscatter value that demarcates water from not-water pixels [47]. However, identifying an effective threshold is affected by the shape of the backscatter histogram, determined by the proportion of open water and land covers within the mapping area [48] and by the waterbody's surface roughness at the time of the acquisition [49]. Water surface roughness and backscatter will vary with wind conditions and, when present, increase with increasing emergent vegetation amount and height (e.g., paddy fields, vegetated water edges, floating weeds) [48]. As the number and size of waterbodies within the landscape decreases, the histogram will shift from bi-modal to unimodal, with the threshold value at the bottom and lower value end of this single peak [50] (Fig 4). Increases in backscatter variability results in a threshold shift towards higher values.

**Available approaches.**   There are automated approaches for thresholding imagery, such as the Otsu thresholding [48, 51], minimum error thresholding [52], the valley emphasis method [50] or Bayesian approaches, e.g., [35, 46, 53, 54]. Otsu and minimum error thresholding, also used for flood mapping, e.g., [47, 55], require a bimodal histogram and so were unsuitable in the case of our three districts which showed unimodal histograms during the drier months of the year (Fig 4B). Also, when we trialled Otsu for a smaller area showing a bimodal histogram

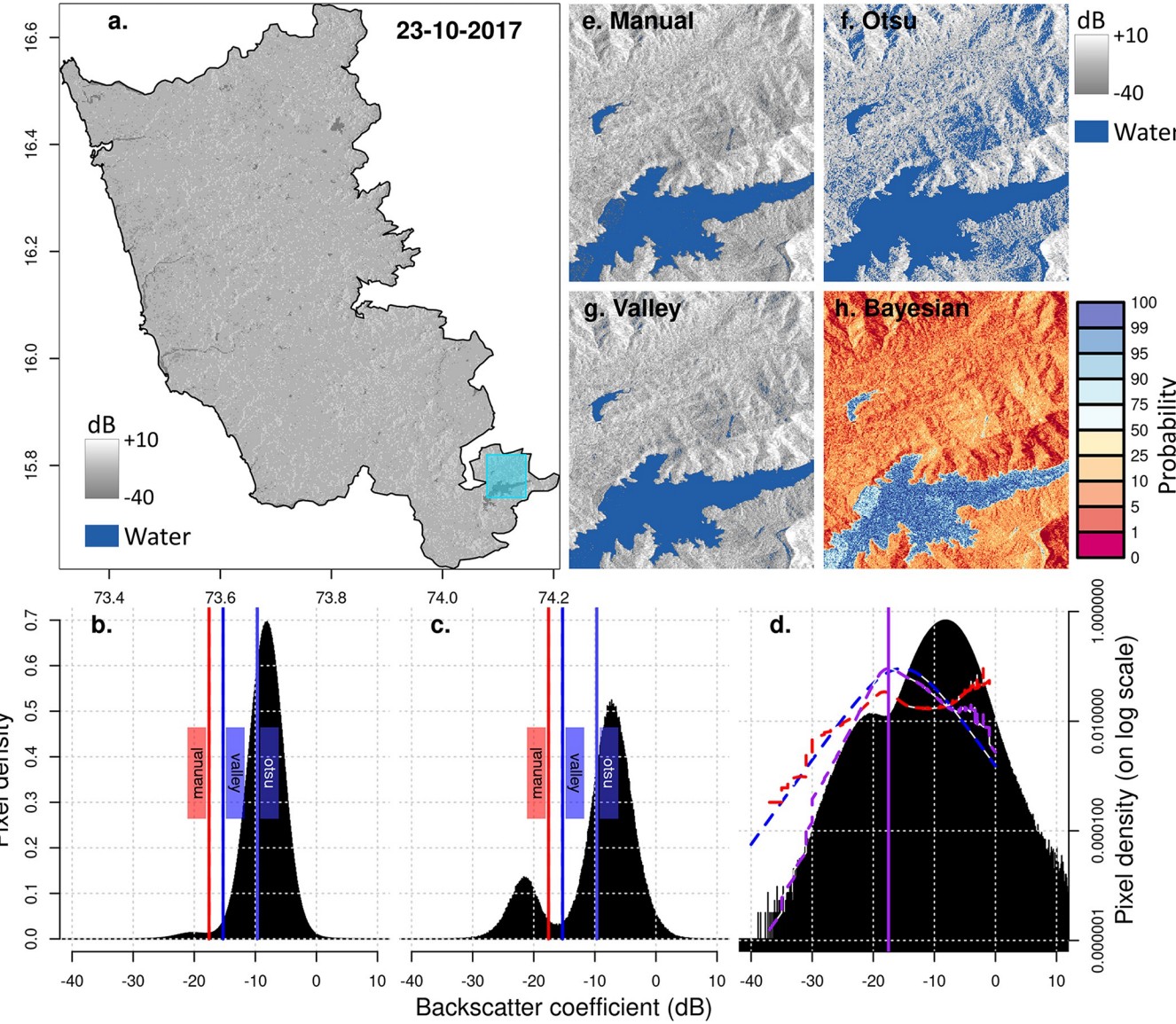

**Fig 4. Example thresholds.** (a) Example Sentinel-1A SAR VV backscatter image for October 2017, Sindhudurg. Source of imagery: Copernicus Sentinel-1 data 2017 accessible via https://dataspace.copernicus.eu/explore-data/data-collections/sentinel-data/sentinel-1 or through GEE using ee.ImageCollection ("COPERNICUS/S1_GRD")); (b) Unimodal backscatter histogram for the whole image covering Sindhudurg district; (c) Bimodal backscatter histogram for the highlighted area in the Sentinel-1A SAR image. Red, blue and dashed blue vertical lines are threshold values obtained using manual, valley and Otsu, respectively. (d) Bimodal backscatter histogram on a log scale with the Bayesian inferred likelihood of a backscattering coefficient value being a suitable threshold (purple dashed line) and the most likely threshold (vertical solid purple line). We decompose the likelihood into prior probability (blue dashed line) and posteriori probability (red dashed line) (see S1 Appendix). (e-f) Resultant waterbody area (in blue) using threshold derived from the whole image shown for the highlighted area using the (e) manual thresholding (threshold: -17.58 dB); (f) Otsu thresholding (threshold: -8.76 dB), (g) Valley Emphasis approach (-15.28 dB). (h) The probability of being a waterbody using the Bayesian approach. Areas in blue are most likely waterbodies, while areas in red are least likely waterbodies.

(Fig 4C) the resulting threshold value mapped too much water (Fig 4E). Based on Otsu thresholding, the valley emphasis method is designed to find the threshold for both unimodal and bimodal histograms by giving more weight to the histogram valley points. However, this approach remains less accurate when the object's (i.e., surface water area) backscatter variance is very different from the background [32]. Fan and Lei [56] trialled the valley emphasis method, the Otsu method and manual thresholding on a landscape in Greece, including wetlands and paddy fields, and found that the valley emphasis method outperformed the Otsu method and matched manual thresholding. Bayesian approaches, sometimes used for flood mapping, derive a threshold from two overlapping Gaussian [35, 54] or Gamma [53] distribution functions fitted to the image backscatter histogram. All require a bimodal distribution and a threshold prior. To guarantee a bimodal distribution, whole SAR images are split into fixed [35] or changeable tile sizes [53], and thresholds are only derived from tiles with a bimodal distribution. Approaches avoid using a non-informative prior (i.e. both water and non-water are equally likely, see [57]) and instead determine it manually by implementing Otsu or using independent flood or water maps. More recently, Chen et al. [58] proposed an adaptive thresholding approach applied on all tiles, automatically identifying unimodal and bimodal histograms to generate different thresholds. The tile histogram shape is determined a priori by mapping persistent water bodies using 16 years of optical (Landsat) data for the same region.

**Approach selection.** Here (A in Fig 1), we implemented the Otsu method (Fig 4F), the valley emphasis method (Fig 4G, see S1 Appendix and S1 Table), manual thresholding (Fig 4E) and a Bayesian inference approach (Fig 4H). For the manual approach, we identified the threshold through trial and error guided by Landsat ETM-based land cover maps produced for 2017 [59], identifying a value that avoided mapping too many single pixels as water. The Bayesian method was developed to analyse SAR imagery for an entire district regardless of the backscatter distribution (unimodal or bimodal). This method estimates the probability of each backscatter value being an effective threshold for detecting waterbodies. It begins with a prior distribution calculated from reference backscatter values for water and non-water (S1 Fig). It then incorporates likelihoods based on image noise, total waterbody area, and backscatter histogram valley magnitude. For image noise, we assume that waterbodies are typically connected (i.e., cover more than one or two image pixels) and noisy areas in the images are less likely to be water. The total waterbody area assumes that there is generally more land than water in an image. Finally, if the threshold falls within a valley on a histogram, it's more likely to be an effective threshold. Further details about our approach, how these likelihoods are combined, and the sampling methods can be found in S1Appendix. Method performance was compared using in situ waterbody /not-waterbody reference points (see accuracy assessment section and results in S2 Table).

The valley emphasis method performed better than Otsu (Fig 4) but it still required manual tuning (see S1 Appendix). Also, because we are dealing withflood events in Shivamogga and Wayanad and varied landscapes across the three very large study areas, we found its performance was varied and sometimes inferior to our manual thresholding (Table 2, S2 Table). In contrast, the most likely thresholds identified by the Bayesian approach were very similar to the manual thresholds, except during the dry season when there are fewer waterbodies, and during the monsoon season when there are more small waterbodies (Table 2). When Bayesian and manual thresholds are similar (Table 2) the Bayesian most likely threshold achieved slightly higher Users and Producers accuracies (see test for Sindhudurg post-monsoon image–S2 Table). Still, although the high accuracy was promising, without in-situ data collected in the pre-monsoon and monsoon period for the same region we were not able to evaluate its performance across seasons. The Bayesian approach is designed to avoid small waterbodies (i.e., they

**Table 2. For Sindhudurg Sentinel-1A SAR images, the most likely thresholds identified using valley emphasis (bin25) and Bayesian inference compared with our corresponding manual thresholds.**

| Date | Bayesian inference * most likely (10%, 90%) (dB) | Valley Emphasis with bin 25 (dB) | Manual threshold (dB) | Difference between manual and Valley Emphasis / Bayesian (dB) | Manual threshold probability ** | Valley Emphasis probability ** |
|---|---|---|---|---|---|---|
| 09/03/ 2017 [+] | -16.11 (-19.89, -8.11) | -17.44 | -17.61 | -0.17 / -1.51 | 97.24 | 97.96 |
| 25/06/ 2017 [++] | -17.09 (-20.09, -9.14) | -14.49 | -17.33 | -2.84 / -0.24 | 99.98 | 91.43 |
| 17/09/ 2017 [&] | -15.90 (-19.89, -8.11) | -15.64 | -15.49 | +0.15 / +0.41 | 99.80 | 99.90 |
| 23/10/ 2017 [&&] | -17.51 (-20.15, -7.33) | -15.28 | -17.58 | -2.3 / -0.07 | 100 | 94.64 |
| 10/12/ 2017 [&&] | -16.11 (-19.78, -6.19) | -16.09 | -16.33 | -0.24 / -0.22 | 99.99 | 99.99 |

[+] Dry season; [++] Start of Monsoon; [&] Monsoon; [&&] Post monsoon and in-situ data available for validation.

\* Based on [60]

\*\* probability of the proposed threshold being effective in detecting waterbodies

would appear as "noise" in the algorithm). This can bias the threshold identification (threshold in monsoon for Bayesian is lower than for manual in Table 2). Also, flooded paddies have backscatter values that are higher than open water (but lower than non-flooded areas) when the rice crop is planted and grows (S1 Fig). Thus, flooded paddy fields and other small seasonal water features are likely to be missed during the monsoon, leading to an underestimation of water body number and extent. To prevent this potential bias impacting the comparison with the JRC product we choose to implement the manual approach, even if it may not perform as well as the automated Bayesian approach.

We applied a standard average threshold to all months except the monsoon months (Table 3). This was because manual thresholds were very similar for each district and consistent across time, except during the monsoon season when the threshold increased in value to allow for flooded paddy fields (i.e., from average values of -16.97 dB to -17.97 dB to values between -13.36 dB and -15.65 dB, in Wayanad in July 2017 and June, July and August 2018; in Shivamogga and Sindhudurg in July 2017 and 2018 –see S4 Table). Although paddy fields, when flooded, have lower backscatter, this backscatter is generally higher than open waterbody backscatter [43, 61]. We also found that during the monsoon season, some waterbody backscatter increases significantly (S1 Fig). It is unclear why, but we suspect it may be due to an increase in water surface roughness caused by stormy winds.

**Table 3. Manual thresholding: The district's average backscatter threshold values (and standard deviation) for 2017 and 2018.**

| Year | Region | Average Threshold, StDev (dB) |
|---|---|---|
| 2017 | Shivamogga | -17.49, 0.38 |
| 2018 | Shivamogga | -16.97, 0.80 |
| 2017 | Sindhudurg | -17.37, 0.65 |
| 2018 | Sindhudurg | -17.85, 0.68 |
| 2017 | Wayanad | -17.97, 0.72 |
| 2018 | Wayanad | -17.75, 1.21 |

Sentinel-1A's temporal resolution is 12 days, leading to two available acquisitions per month. To produce a monthly surface water area map, we combined the surface water pixels identified in one or the other image (i.e., a pixel was mapped as 'water' when it was identified as water in either image). We refer to our monthly surface water area maps for 2017 and 2018 as 'S1A' hereafter.

**Accuracy assessment.** To evaluate the accuracy of the S1A and JRC surface water maps, we compared both against 'water' and 'non-water' observations collected in the field in August 2018, December 2017, and November 2018 for Shivamogga, Sindhudurg and Wayanad, respectively. The reference points were collected as part of a related, but independent land cover mapping exercise of the three districts [59]. Ideally, a spatially stratified random approach would ensure that reference points are unbiased and represent the relative abundance of cover classes [62]. However, in reality, this was not achievable and instead, similar to other large-scale mapping exercises, e.g., [63], we collected points along roads or accessible paths. This was especially the case for Shivamogga, where extreme flooding across the district at the time of the survey limited point collection along major road arteries. We used a total of 3202 reference points: 890 in Shivamogga, 705 in Sindhudurg and 1606 in Wayanad (Fig 5).

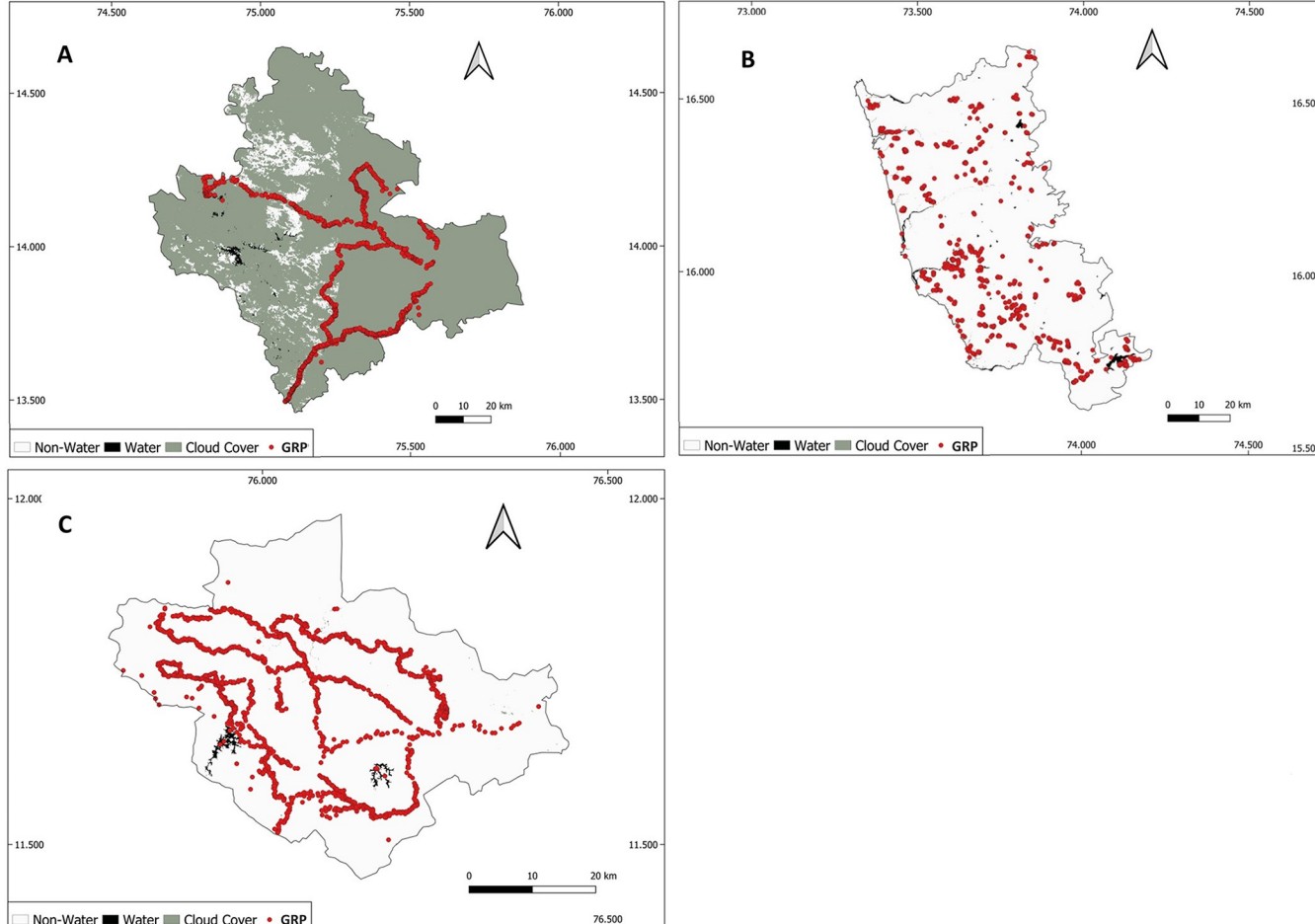

**Fig 5.** Distribution of ground reference points (GFP) displayed with the S1A water body map and JRC cloud cover layer for the month matching the timing when reference points were collected in situ:A. Shivamogga, B. Sindhudurg and C. Wayanad. Source of Administrative boundaries: The Global Administrative Unit Layers (GAUL) dataset, implemented by FAO within the CountrySTAT and Agricultural Market Information System (AMIS) projects.

'Water' reference points included lakes, rivers, ponds, reservoirs, streams, and flooded paddy fields (i.e., 157 points are paddy fields). 'Non–water' reference points included built-up, forest, grassland, plantation, fallow land, and agricultural land.

We evaluated the mapping accuracy through a percentage area-based confusion matrix as per [62] for the monthly surface water area maps that match the field survey month and year. We calculated overall producer(i.e., 1—omission error) and user(i.e., 1—commission error) accuracy and their associated standard error. For both S1A and JRC maps, we masked out the reference points that, according to the JRC map's quality tag, were obscured by cloud during the field data collection (i.e., we removed 859 reference points for Shivamogga and none for Sindhudurg and Wayanad).

### Comparison with JRC global surface water product

For a spatial comparison between our S1A and the JRC map, we calculated the following set of land metrics per district [64]:

- Total area (km$^2$) of 'water', 'non-water' and JRC 'cloud' mapped across a district.

- Patch number (PN) is the total number of surface water areas within a district.

- Mean patch size (MPA), the number of surface water areas within a district divided by the total surface water area.

- Patch density (PD), the number of surface water areas within a district divided by total district area (m$^2$) (multiplied by 10,000 to simplify plots).

To compare patch number, mean patch size and patch density, we eliminated the impact of cloud cover by deriving the metrics for the months when the JRC product has a minimum cloud cover ($< 20$ km$^2$) and excluded these cloudy areas from our surface water area map. We generated the landscape metrics over a land area of 8,445 km$^2$ for Shivamogga, 5,171 km$^2$ for Sindhudurg and 2,122km$^2$ for Wayanad.

### Results

Due to the slightly higher spatial resolution of the SAR data (i.e., ~10m versus 30m) and the sensitivity of microwaves to waterbodies, S1A identifies more and smaller waterbodies. Fig 6 shows examples of the S1A (left) and the corresponding JRC waterbody maps (middle). It also includes flooded rice paddies, which JRC does not map. The impact of cloud cover on JRC (Fig 6–right) is substantial during the rainy season (generally from May to September) when clouds hide large areas of the study sites. For example, in Shivamogga in July 2017, cloud covered 8420.982 km$^2$, representing nearly the entire district area. JRC is also affected byscan line correction gaps visible in the cloud mask of Sindhudurg and Wayanad (Fig 6–right). This is a known issue with the Landsat7 sensor [65].

For Shivamogga, we could not calculate mapping accuracies as clouds obscured nearly allreference points. For the remaining districts (Sindhudurg and Wayanad), the overall map accuracy is consistently high for both maps (Table 4). Both maps' user and producer accuracies vary between districts and the water and non-water classes. For water, the S1A map consistently achieves relatively high producer (low omission error) and user (low commission error) accuracies compared to the JRC map (Table 4). The water producer accuracies for S1A are higher than the user accuracies for Wayanad (i.e., commission error is higher than omission error). In contrast, it is the opposite for JRC (user accuracy is higher than producers', or there are more omission errors). For non-water, both maps show high producer and user accuracies.

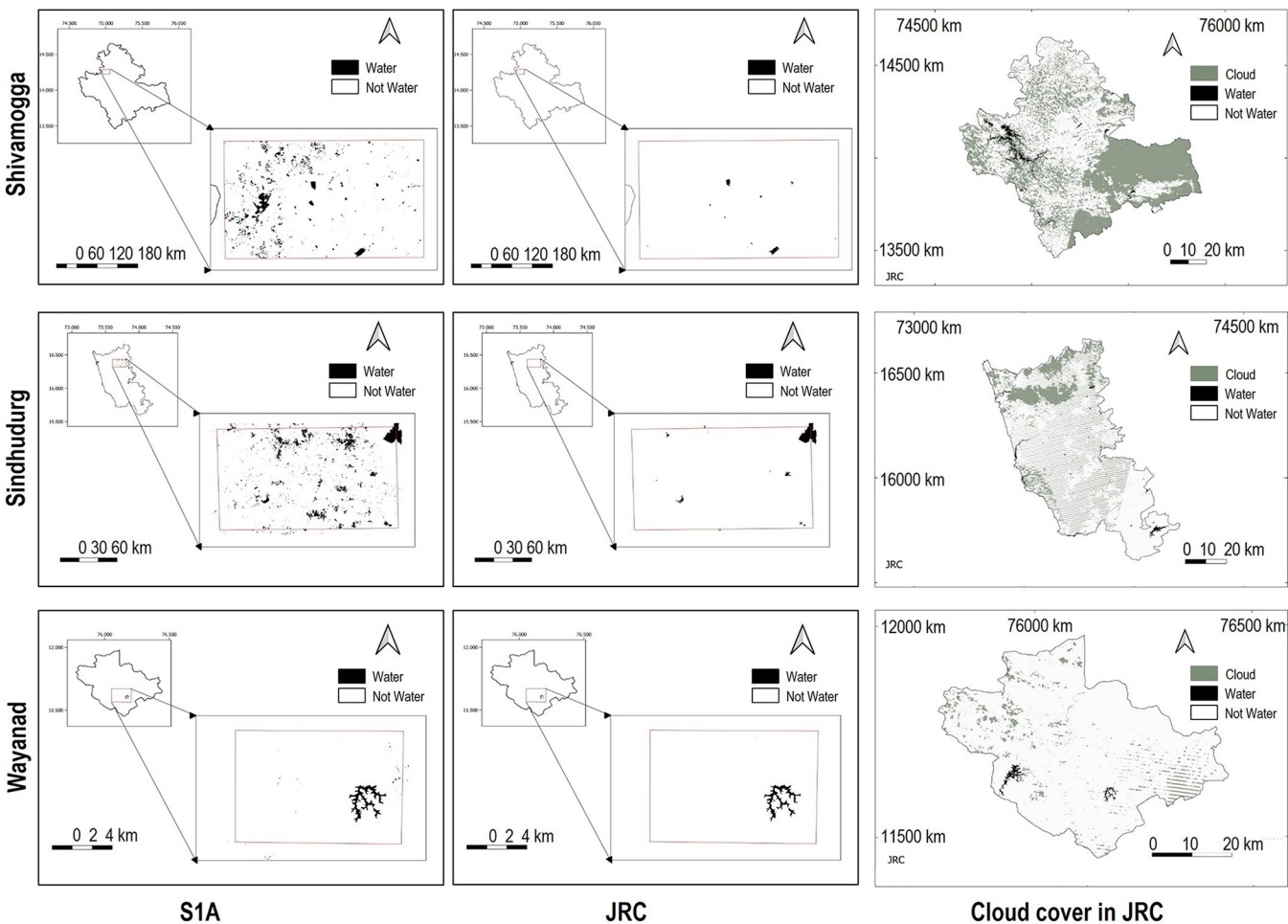

**Fig 6. Surface water area and cloud cover impact in Shivamogga, Sindhudurg and Wayanad for January 2018.** S1A (left) and JRC (middle) surface water area (shown in black); and for JRC (right), the impact of cloud cover (shown in grey) and scan line correction gaps. White represents areas within the districts where there is no surface water.

When comparing the S1A mapping performance between districts, we see a lower user accuracy (or higher commission error) for Sindhudurg (i.e., 90.45%) and a low producer accuracy (or high omission error) for Wayanad (i.e., 79.9%). The commission error in Sindhudurg

**Table 4. Accuracy assessment of waterbodies in Shivamogga, Sindhudurg and Wayanad showing average user and producer accuracy and standard error values for S1A and JRC-derived surface water maps and the number of water (W) and non-water (NW) reference points.**

| District | Date | Data | Water<br>% UA (SE) | Water<br>% PA (SE) | Non-Water<br>% UA (SE) | Non-Water<br>% PA(SE) | % OA (SE) | W / NW<br>N° Ref points |
|---|---|---|---|---|---|---|---|---|
| Shivamogga | Aug-18 | S1A | 91.1 (2.3) | 92.4 (3.1) | 99.3 (0.3) | 99.2 (0.2) | 98.6 (0.34) | 3 / 28 |
| | | JRC | 0 (0) | 0 (0) | 100 (0) | 8.1 (2.6) | 14.2 (0.08) | 3 / 28 |
| Sindhudurg | Dec-17 | S1A | 90.45 (0.67) | 97.12 (1.06) | 99.9 (0.01) | 99.9 (0.09) | 99.6 (0.12) | 47 / 659 |
| | | JRC | 100.0 (0) | 49.3 (8.8) | 98.8 (0.4) | 100 (0) | 98.6 (2.43) | 47 / 659 |
| Wayanad | Nov-18 | S1A | 100.0 (0) | 79.9 (9.2) | 99.8 (0.1) | 100 (0) | 99.8 (0.40) | 30 / 1576 |
| | | JRC | 100.0 (0) | 35.7 (9.3) | 98.8 (0.5) | 100 (0) | 98.8 (0.46) | 30 / 1576 |

UA: User Accuracy; PA: Producer Accuracy; OA: Overall Accuracy; W: Water; NW: Non-Water

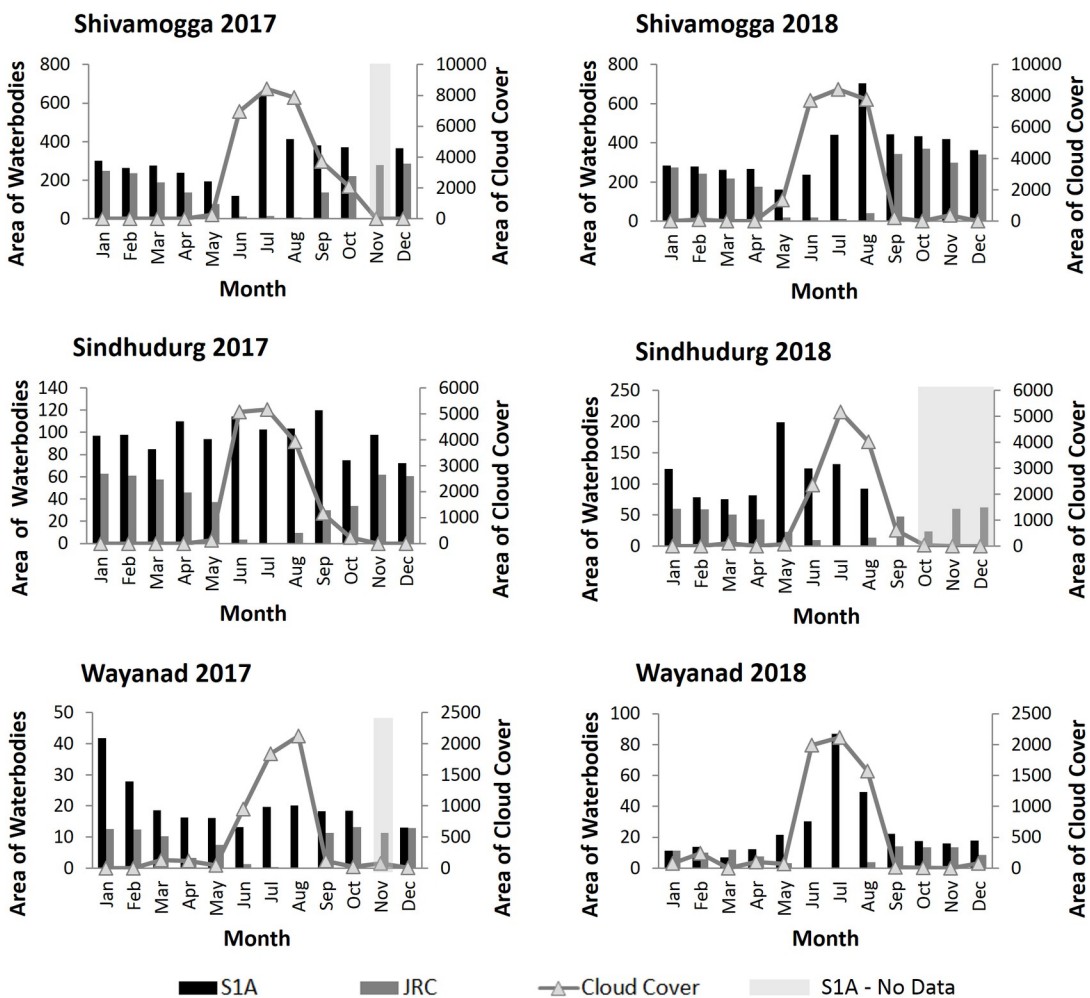

**Fig 7. Seasonal variation in the total surface water and cloud cover area derived from the S1A and JRC waterbody maps for Shivamogga, Sindhudurg and Wayanad, 2017 and 2018.** Across the districts, the annual average is 25% for 2017 and 23% for 2018, while the average maximum cloud cover in July or August is 100% for both 2017 and 2018.

appears to be due to radar shadow caused by steep mountainous terrain. In Wayanad, the higher cases of omission are primarily associated with river channels, which we did not map during winter due to high canopy cover obscuring the water.

Across all months and districts, the total surface water area mapped by S1A is higher than JRC (Fig 7). However, the difference is slight in December and January when cloud cover is minimal; for these months, the average area difference is 1.56 km$^2$ in Shivamogga, 0.281 km$^2$ in Sindhudurg and 2.80 km$^2$ in Wayanad. During the monsoon season, between June and August, when cloud cover is high, JRC detects almost no waterbodies, while S1A shows a seasonal peak across all three districts (Fig 7). The June to August average monthly area difference between S1A and JRC is 7855.35 km$^2$ in Shivamogga, 4288.84 km$^2$ in Sindhudurg and 1764.606 km$^2$ in Wayanad.

Patch number and density are consistently higher in S1A than JRC and the mean patch area is smaller (Fig 8). On average S1A identifies 34894, 17554 and 2785 more patches in Shivamogga, Sindhudurg and Wayanad, respectively, the majority of which are small throughout the year. This is because of the Sentinel-1A SAR 's slightly higher spatial resolution and the strong specular reflectance over water surfaces, which helps identify very small waterbodies

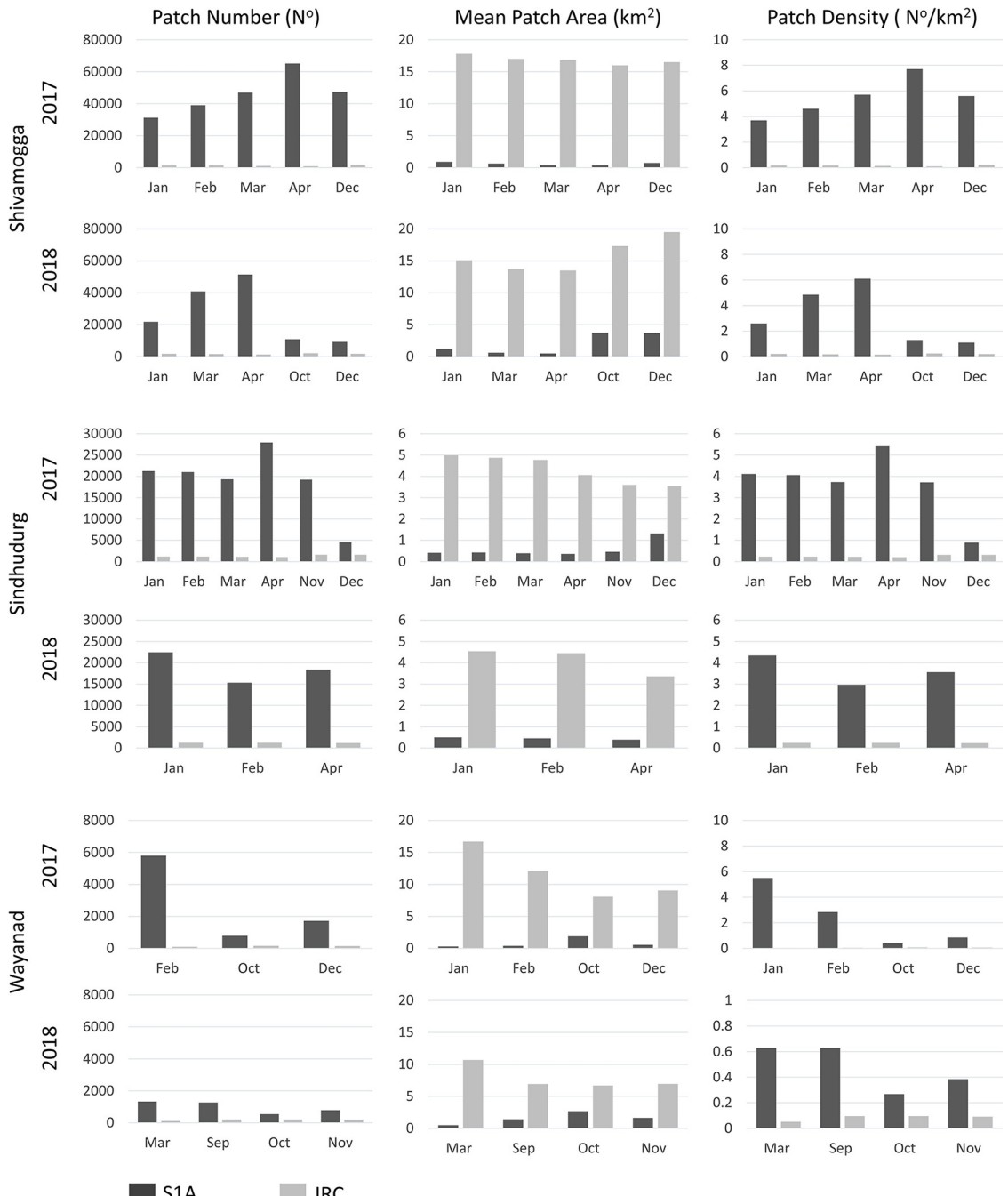

**Fig 8. District surface water area patch number, mean patch area and patch density derived from the S1A and JRC maps for months when JRC has the least cloud cover (<20%).**

and flooded paddy areas from other land surface covers. About 91% of our 157 paddy reference points were mapped as water.

## Discussion

Accurate and high temporal and spatial resolution surface water mapping is vital for modelling the risks of wide-ranging water-related diseases. It is also relevant to other applications

requiring seasonal mapping, such as, monitoring water availability in water scares regions, or quantifying the contribution of seasonal and/or small lakes as carbon emitters [66].

Here we show that radar-based mapping is essential for mapping small water bodies in regions that experience dense seasonal cloud cover and have dynamic surface water coverage. Sentinel-1A SAR is unaffected by clouds and provides high frequency and high spatial resolution data matched to the seasonal disease transmission dynamics as well as local scales of human pathogen exposure and interventions. Using a simple manual thresholding approach, we mapped surface water availability more accurately with the Sentinel-1A SAR time-series than with the optically based JRC global surface water product (see Table 4). Moreover, while both S1A and JRC achieved high overall accuracies (> 98%) when the cloud cover was low, the extent of unmapped surface water area in the JRC product was substantial during the cloudy monsoon months. This is the case for all three districts, even though they have varied landscapes with different land cover, land use, hydrology, and seasonal rainfall patterns (Fig 7). During the dry (cloud-free) season the difference in total water area detected between S1A and JRC is relatively small, however, JRC's large water omission errors (i.e., 50.7% and 64.3% for Sindhudurg and Wayanad respectively) suggests that the Sentinel-1A SAR higher spatial resolution helped detect many more smaller sized waterbodies (including flooded paddy fields) (Fig 8). According to our social scientists conducting community household surveys in the landscape, people, livestock and wildlife, all of which are hosts for pathogens and vectors, use waterbodies as small or smaller than $10m^2$ as well as larger water-body edges for watering. So, capturing the complete range of water-body sizes is key to understand environmental transmission. The use of higher spatial resolution optical imagery (e.g., from Sentinel-2 or Planet Labs) could resolve this, however where disease risks are linked to the rainy season, and transmission is linked to small waterbodies, the benefits of using high spatial resolution SAR over optical for surface water mapping are particularly pronounced, especially in tropical contexts where rainy seasons are associated with dense cloud cover. Even where the high-risk periods for disease coincides with the dry period following the rainy season, SAR may still be preferable for disease mapping since the extent and persistence of exposure to disease may depend on the waterbodies present during the prior rainy season [67]. More-over for many water-related diseases (e.g., Rift Valley Fever Virus), the sequence of drought followed by a flood can amplify transmission by concentrating hosts, vectors, and people within the remaining waterbodies in the landscape. Given the loss of cloud free optical data during the rainy season in tropical environments, it would be better to track such important sequences and predict consequent seasonal transmission risks using SAR.

C band radar is particularly good at mapping open surface water, as evidenced by a variety of studies, e.g., [14, 31, 32, 54, 55, 68], but less effective at mapping vegetated surface waters, narrow rivers with overhanging vegetated on their banks or wetlands [55]. Flooding is also not visible under thick forest cover [69]. Still, we could map fully submerged rainfed paddy fields with low vegetation cover (91% of our 157 paddy reference points were mapped as water during the monsoon). This aligns with a study in Africa [14], which successfully identified surface water areas in with < 20 cm tall and spaced-out grasses using Sentinel-1 SAR.

We have shown that cloud cover can substantially impact JRC's dynamic surface water mapping (Figs 7 and 8). Still, because of the lack of suitable radar alternatives, most efforts, prior to the Sentinel-1 launch in 2014 (A) and 2016 (B), have relied on the Landsat optical sensor series (1980s-to date). Recently, the 30m global flood monitoring system (GFM) (https://extwiki.eodc.eu/en/GFM) has been using Sentinel-1A SAR to map flood events in real-time as SAR data is acquired. It also maps waterbodies that are permanently or seasonally present in the 2 years preceding a flood year. However, until then, the JRC product was the only global operational product that delivered monthly surface water maps. As a result, JRC has also been

used to train, e.g., [14] or inform, e.g., [33] more recent and alternative surface water mapping approaches, introducing a real risk of error propagation, especially where seasonal surface water appears during rainy (and cloudy) seasons. For example, when using JRC as a training data set, the 'open water' class [14] requires 12 months of JRC detection. Using SAR based time-series instead would drastically increase the number of open water training samples in areas with typical monsoon periods, improving classification performance. Finally, some surfaces have similar backscatter to open water, such as snow or sandy areas [70]. These were absent in our study areas and so in our case did not require special attention.

Although our manual thresholding approach proved effective at the local scale, upscaling to national or continental scales may require automation. In some instances, applying the Valley emphasis method can be as effective as manual thresholding, e.g., [32], and Otsu thresholding has proven useful as a post-correction [14]. Both should be described as semi-automated as they both require a manual tuning of the histogram bins. The currently preferred approach for automation is to split whole SAR images into tiles to achieve bi-modal backscatter distributions. Next, using Bayesian modelling, tile thresholds are identified and subsequently averaged to produce a single image threshold. For example, the GFM system uses 3 independent Bayesian-based algorithms, two of which implement image tiling [54, 57, 71, 72]. Alternatively [58] developed a tile thresholding that automatically adapts to the tile's histogram shape (uni- or bi-modal) which is determined a priori from mapping multiple years of optical data (i.e., 16 years of Landsat). While the reported high accuracies are encouraging, map evaluation was limited to a single month and does not provide any insights into how cloudy seasons may impact the a priori data and ensuing final map. In our case, substantial changes in threshold values during the monsoon season suggest that successful automation requires a geographically flexible approach, or one that is at least flexible in dynamic areas with large seasonal or periodic changes in open water and flooded paddies (or other sparse vegetation).

Like other studies, we used Bayesian inference to inform the likelihood of a backscatter value being a suitable threshold. These likelihoods are determined by manually identified priors, informing the algorithm where in parameter space to search (i.e., within a minimum and maximum backscatter value–S1 Fig), and by applying rules. In our case the rule was to quantify backscatter image noise, assuming waterbodies are less noisy (see methods). Ensuring flooded paddy fields are identified as waterbodies is more complex as the backscatter changes seasonally when paddy fields are flooded, and the rice crop grows. This will require us to develop additional rules.

Uniquely, we can also use Bayesian inference to establish how closely manual thresholds are to the most likely threshold (i.e., manual threshold probability in Table 2). However, Bayesian methods are most effective when the full likelihoods of all thresholds are considered. This, similar to [73] enables a spatially explicit confidence map for (i) waterbodies, through uncertainty estimates of the plausible range of thresholds and (ii) disease risk model outputs by propagating the waterbody confidence maps through to the models. For health preparedness and response, propagating uncertainty enables the modelling of disease risks, arising from likely-, least- and worst-case scenarios of extent, and associated waterbody change impacts.

## Conclusions

To study, monitor and forecast infectious diseases risks linked to variability in surface water, we require consistent spatial and temporal data on surface water dynamics across large regions. Even though the 30m JRC Global Surface Water product achieves high surface water mapping accuracies when there is no cloud, using Sentinel-1A SAR data becomes essential for areas with highly dynamic surface waters and cloudy/rainy seasons as the impact of cloud

cover on optically derived seasonal information is substantial. Processing of Sentinel-1A SAR 's more detailed 10m spatial resolution data, helped detect many small water features missed by the 30m JRC Global Surface Water product, especially during the cloudy monsoon season. Backscatter image thresholding proved very effective to map unvegetated surface waterbodies, but only if threshold values are adapted to regional-specific spatial and temporal variations in backscatter. Applying Bayesian image thresholding onto Sentinel-SAR, if tuned to detect small and seasonal waterbodies, may prove the best automated option for mapping waterbody extent in tropical rainy landscapes. Still, without meter resolution SAR imaging, many very small water bodies, relevant for disease risk modelling and other applications, will remain unmapped.

## Supporting information

**S1 Appendix. Bayesian inference method and valley emphasis method.**
(DOCX)

**S1 Fig. Sentinel1-SAR VV backscatter time-series for various land cover features in Sindhudurg.**
(TIF)

**S1 Table. Threshold values derived using the valley emphasis method for Sindhudurg Sentinel 1 SAR 2017 images, selected to represent the dry (March), monsoon (June-September) and post-monsoon (October-December) seasons.**
(DOCX)

**S2 Table. Waterbody mapping accuracies achieved using the manual, Bayesian inference and Valley emphasis thresholding approaches for the Sindhudurg Sentinel-1A SAR image of 10th December 2017.**
(DOCX)

**S3 Table. Manual thresholds for each Sentinel-1 SAR image.** The threshold values highlighted in yellow are when largescale flooding occurred.
(DOCX)

**S4 Table. Total waterbody area per district as estimated from S1A and JRC.** Total cloud cover area as estimated from JRC.
(DOCX)

**S5 Table. Per district: Number of patches, mean patch area and patch density calculated from S1A and JRC waterbody maps.**
(DOCX)

## Acknowledgments

We thank government organizations for providing permission to conduct fieldwork and Dr. Darshan, Abhijit, M Mubashira and Dr. Irfan for helping during Shivamogga field work.

## Author Contributions

**Conceptualization:** France F. Gerard.

**Data curation:** Gowri Uday, Abhishek Samrat, Anusha Chaudhary, Mujeeb Rahman.

**Formal analysis:** Gowri Uday.

**Funding acquisition:** Bethan V. Purse, Abi Vanak.

**Investigation:** Gowri Uday.

**Methodology:** Gowri Uday, Douglas I. Kelley, Abhishek Samrat, France F. Gerard.

**Project administration:** Bethan V. Purse, Abi Vanak.

**Resources:** Bethan V. Purse, Abi Vanak.

**Software:** Gowri Uday, Douglas I. Kelley, Abhishek Samrat.

**Supervision:** France F. Gerard.

**Validation:** Bethan V. Purse, Douglas I. Kelley, France F. Gerard.

**Visualization:** Gowri Uday, Douglas I. Kelley, France F. Gerard.

**Writing – original draft:** Gowri Uday.

**Writing – review & editing:** Bethan V. Purse, Douglas I. Kelley, Abi Vanak, France F. Gerard.

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
