## [Decision Letter · Decision Letter 0]

29 Jan 2024

PONE-D-23-35272In monsoon-affected India, Satellite radar provide a fine-grained understanding of seasonal surface water availability for health applicationsPLOS ONE

Dear Dr. France,

Thank you for submitting your manuscript to PLOS ONE. After careful consideration, we feel that it has merit but does not fully meet PLOS ONE’s publication criteria as it currently stands. Therefore, we invite you to submit a revised version of the manuscript that addresses the points raised during the review process.

We look forward to receiving your revised manuscript.

Kind regards,

Mohammed Sarfaraz Gani Adnan, PhD

Academic Editor

PLOS ONE

 [The MonkeyFeverRisk project that led to these results is supported by the Global Challenges Research Fund and funded by the MRC https://www.ukri.org/councils/mrc/, AHRC https://www.ukri.org/councils/ahrc/, BBSRC https://www.ukri.org/councils/bbsrc/, ESRC https://www.ukri.org/councils/esrc/ and NERC https://www.ukri.org/councils/nerc/ [grant numbers MR/ P024335/1 and MR/P024335/2], awarded to BVP, AV and FG. Additional support was provided from the IndiaZooRisk Project, which is funded by UK Research and Innovation https://www.ukri.org/councils/nerc/ through the Global Challenges Research Fund [MR/T029846/1] and by the NERC SUNRISE project [grant number NE/R000131/1]. DIK was supported by the Natural Environment Research Council as part of the NC-International programme [NE/X006247/1] delivering National Capability. ].  

[The MonkeyFeverRisk project that led to these results is supported by the Global Challenges Research Fund and funded by the MRC, AHRC, BBSRC, ESRC and NERC [grant numbers MR/ P024335/1 and MR/P024335/2], awarded to BVP, AV and FG. Additional support was provided from the IndiaZooRisk Project, which is funded by UK Research and Innovation through the Global Challenges Research Fund [MR/T029846/1] and by the NERC SUNRISE project [grant number NE/R000131/1]. DIK was supported by the Natural Environment Research Council as part of the NC-International programme [NE/X006247/1] delivering National Capability. 

We thank government organizations for providing permission to conduct fieldwork and Dr. Darshan, Abhijit, M Mubashira and Dr. Irfan for helping during Shivamogga field work.]

  [The MonkeyFeverRisk project that led to these results is supported by the Global Challenges Research Fund and funded by the MRC https://www.ukri.org/councils/mrc/, AHRC https://www.ukri.org/councils/ahrc/, BBSRC https://www.ukri.org/councils/bbsrc/, ESRC https://www.ukri.org/councils/esrc/ and NERC https://www.ukri.org/councils/nerc/ [grant numbers MR/ P024335/1 and MR/P024335/2], awarded to BVP, AV and FG. Additional support was provided from the IndiaZooRisk Project, which is funded by UK Research and Innovation https://www.ukri.org/councils/nerc/ through the Global Challenges Research Fund [MR/T029846/1] and by the NERC SUNRISE project [grant number NE/R000131/1]. DIK was supported by the Natural Environment Research Council as part of the NC-International programme [NE/X006247/1] delivering National Capability. ].

6. Please amend either the title on the online submission form (via Edit Submission) or the title in the manuscript so that they are identical.

7. We note that Figure(s) 1, 3, 4 and 5 in your submission contain [map/satellite] images which may be copyrighted. All PLOS content is published under the Creative Commons Attribution License (CC BY 4.0), which means that the manuscript, images, and Supporting Information files will be freely available online, and any third party is permitted to access, download, copy, distribute, and use these materials in any way, even commercially, with proper attribution. For these reasons, we cannot publish previously copyrighted maps or satellite images created using proprietary data, such as Google software (Google Maps, Street View, and Earth). For more information, see our copyright guidelines: http://journals.plos.org/plosone/s/licenses-and-copyright.

a. You may seek permission from the original copyright holder of Figure(s) 1, 3, 4 and 5 to publish the content specifically under the CC BY 4.0 license.  

8. We note that Figure 2 in your submission contain copyrighted images. All PLOS content is published under the Creative Commons Attribution License (CC BY 4.0), which means that the manuscript, images, and Supporting Information files will be freely available online, and any third party is permitted to access, download, copy, distribute, and use these materials in any way, even commercially, with proper attribution. For more information, see our copyright guidelines: http://journals.plos.org/plosone/s/licenses-and-copyright.

Reviewers' comments:

Reviewer's Responses to Questions

**Comments to the Author**

1. Is the manuscript technically sound, and do the data support the conclusions?

Reviewer #1: Partly

Reviewer #2: Partly

2. Has the statistical analysis been performed appropriately and rigorously? 

Reviewer #1: N/A

Reviewer #2: Yes

3. Have the authors made all data underlying the findings in their manuscript fully available?

Reviewer #1: Yes

Reviewer #2: Yes

4. Is the manuscript presented in an intelligible fashion and written in standard English?

Reviewer #1: Yes

Reviewer #2: Yes

5. Review Comments to the Author

Reviewer #1: General comments

In this study, Sentinel-1 SAR data were used to map surface water areas using a Bayesian noise recognition algorithm. Results were compared with the optical Joint Research Centre (JRC) Global Surface Water product. The differences between the results from this study and several thresholding methods (e.g., manual identification) were discussed. The results show that Sentinel-1 SAR is more advantageous in high cloudiness and its higher spatial resolution helps to detect small water features. Overall, the research work is described clearly, however, there are issues that need further clarifications.

1. It’s unclear how the Bayesian noise recognition algorithm was used to map the surface water. There should be a summary of the major contribution or improvement of the proposed algorithm.

2. Although health issues such as water-related diseases are mentioned many times in the title and text, this article only implements water extraction. How to better correlate the extraction results with disease risks?

3. Is there a method flow chart that can provide a more intuitive view of the methods used? The figures are of low resolution in the main content and are difficult to read.

Detailed comments:

Introduction

L121: (< 900m2) should be “m2” .

Methods

Data sources

The source of the validation data was not mentioned.

L197 “… the histogram will shift from bi-modal to unimodal …”, suggest reading following study. It proposed an adaptive thresholding approach, and the bi-model and unimodal were automatically identified to generate different thresholds.

Chen, S.; Huang, W.; Chen, Y.; Feng, M. An Adaptive Thresholding Approach toward Rapid Flood Coverage Extraction from Sentinel-1 SAR Imagery. Remote Sens. 2021, 13, 4899. https://doi.org/10.3390/rs13234899

Extraction of waterbodies

L224: A new Bayesian inference approach is mentioned, which is called “a Bayesian noise recognition algorithm” in the abstract. Is there a unified name for this method?

L227: “in section 2.5” There is no such section, please check whether the expression is correct.

L244: What evidence is there that manual approach is better than the Bayesian approach during the monsoon? Why not use the more accurate method?

L251: The units for the threshold values of backscattering coefficients are not stated in many places in the article. The unit of threshold value, dB, should be added.

L268: The values in the table have no units (e.g., dB, %).

Table 3: What does probability refer to in columns 3 and 4 in Table 3? No obvious explanation is found. The content of the table is arranged in a confusing manner.

Accuracy Assessment

L286: Why are points collected along roads when using Sentinel2-MSI as reference data? What is the distance range between the collection points and roads? Is randomly selecting points a better way?

L319: “km2” There are many such formatting problems.

Reviewer #2: I’m giving my comment on the following article title “Satellite radar data provide a fine-grained understanding of seasonal surface water availability in monsoon-affected areas of India for health applications”

Comment:

1. Title should be rewrite it’s not rich

2. Line 32, it may better use sentinel-1 or Sentinel 1

3. Introduction section is very poor. Please rewrite this within 1.5 to 2 page giving all the relevant information (Novelty, limitation, application) and recent studies.

4. Why the study area is selected? put your statement in study area section.

5. Line 188, (< ~30o) degree?

6. Figure 4b where is legend?

7. All of the figure placed in low regulation and unclear. Make sure their better regulation and readable.

8. Discussion should be rewrite with concise information found from the result.

9. Rewrite the conclusion in a informative way.

10. The title suggests a focus on health issues; therefore, there is a need for greater emphasis on how the work contributes to addressing health concerns..

11. Add some recent citation

12. Enhance the quality of the writing throughout.

6. PLOS authors have the option to publish the peer review history of their article (what does this mean?). If published, this will include your full peer review and any attached files.

Reviewer #1: No

Reviewer #2: No

---

## [Author Response · Author response to Decision Letter 0]

5 Aug 2024

Reply to reviewers’ comments

We thank the reviewers for their valuable comments which have helped us improve our manuscript. 

Reviewer #1: 

1. It’s unclear how the Bayesian noise recognition algorithm was used to map the surface water. There should be a summary of the major contribution or improvement of the proposed algorithm.

This is a new method, and while we describe it extensively in the supplement, we agree that it deserves more description in the main paper. We have therefore improved the paragraph (lines 236 – 246):

“The Bayesian method was developed to analyse SAR imagery for an entire district regardless of the backscatter distribution ( unimodal or bimodal). This method estimates the probability of each backscatter value being an effective threshold for detecting waterbodies. It begins with a prior distribution calculated from reference backscatter values for water and non-water (Fig S1). It then incorporates likelihoods based on image noise, total waterbody area, and backscatter histogram valley magnitude. For image noise, we assume that waterbodies are typically connected (i.e., cover more than one or two image pixels) and noisy areas in the images are less likely to be water. The total waterbody area assumes that there is generally more land than water in an image. Finally, if the threshold falls within a valley on a histogram, it's more likely to be an effective threshold. Further details about our approach, how these likelihoods are combined, and the sampling methods can be found in the supplement.”

2. Although health issues such as water-related diseases are mentioned many times in the title and text, this article only implements water extraction. How to better correlate the extraction results with disease risks? The main scope of the paper was to quantify the impact of cloud cover on optically derived water surface maps in a monsoon region by comparing a radar derived water surface map with a popular optically derived water surface product, and to discuss the implications for disease modelling. Including the radar derived maps into disease risk modelling is not within scope. We have altered the title and text in the abstract and introduction to clarify and shift the focus.

3. Is there a method flow chart that can provide a more intuitive view of the methods used? The figures are of low resolution in the main content and are difficult to read. We have added a schematic that summarises the 3 main steps taken in our approach and explained the steps at the very start of the methods sections (lines 124-128).

Detailed comments:

Introduction

L121: (< 900m2) should be “m2” . Done 

Methods

Data sources

The source of the validation data was not mentioned. We describe the collection of the validation data under ‘ accuracy assessment and since the reference data was collected by the authors there is no ‘ external source’. We have added clarified this by editing the text as follows: “ we compared both against ‘water’ and ‘non-water’ observations collected in the field in August 2018, December 2017, and November 2018 for Shivamogga, Sindhudurg and Wayanad, respectively. The reference points were collected as part of a related, but independent land cover mapping exercise of the three districts (58).” (lines 309- 312). 

L197 “… the histogram will shift from bi-modal to unimodal …”, suggest reading following study. It proposed an adaptive thresholding approach, and the bi-model and unimodal were automatically identified to generate different thresholds. Chen, S.; Huang, W.; Chen, Y.; Feng, M. An Adaptive Thresholding Approach toward Rapid Flood Coverage Extraction from Sentinel-1 SAR Imagery. Remote Sens. 2021, 13, 4899. https://doi.org/10.3390/rs13234899 . We have mentioned the work of Chen et al 2021 in section ‘ Available approaches’ and in our discussion. Lines

Extraction of waterbodies

L224: A new Bayesian inference approach is mentioned, which is called “a Bayesian noise recognition algorithm” in the abstract. Is there a unified name for this method? No there is not, to avoid confusion we have replaced “a Bayesian noise recognition algorithm” with the less specific term “ a Bayesian inference approach”. (line 233)

L227: “in section 2.5” There is no such section, please check whether the expression is correct. Done, we now refer to the “accuracy assessment section”. (Line 247)

L244: What evidence is there that manual approach is better than the Bayesian approach during the monsoon? Why not use the more accurate method? Although the Bayesian method was performing better for the post-monsoon period, without in-situ data collected in the pre-monsoon and monsoon period for the same region we have no way of evaluating its performance across seasons. However, in the monsoon season, paddy fields become a significant source of waterbody expansion. They often are small waterbodies surrounded by non-waterbodies, causing them to appear as "noise" in the Bayesian algorithm. This can bias the threshold identification, leading to an underestimation of water body number and extent. Additionally, flooded paddies have dB values that are higher than open water when the rice crop is planted and grows. To address these issues in any automated thresholding system, including the Bayesian method, adjustments need to be made to avoid biased results during the monsoon. For our comparison with the JRC product, it is important to strive for a non-biased approach (such as our manual approach), even if it may not perform as well as an automated approach. 

We have clarified this in the manuscript. Lines 262-272.

L251: The units for the threshold values of backscattering coefficients are not stated in many places in the article. The unit of threshold value, dB, should be added. Done

L268: The values in the table have no units (e.g., dB, %). Done

Table 3: What does probability refer to in columns 3 and 4 in Table 3? No obvious explanation is found. The content of the table is arranged in a confusing manner. It represents the probability of the proposed threshold being effective in detecting waterbodies. We have included this explanation in the table. We have also rearranged the columns to reduce confusion.

Accuracy Assessment

L286: Why are points collected along roads when using Sentinel2-MSI as reference data? What is the distance range between the collection points and roads? Is randomly selecting points a better way?

The reference points were collected from observations in the field not from interpreting Sentinel-2 imagery. As a result, a random sampling is difficult to implement because of accessibility. We have rewritten parts of the text to make sure this is made clear. Lines 304 – 306.

L319: “km2” There are many such formatting problems. Done

Reviewer #2: 

We thank the reviewer for their comments.

1. Title should be rewrite it’s not rich: We have changed the title to ensure it better represents the scope of the paper: “Radar versus optical: the impact of cloud when mapping seasonal surface water for health applications in monsoon-affected India.”

2. Line 32, it may better use sentinel-1 or Sentinel 1 We replaced Sentinel1 with Sentinel-1 throughout the document.

3. Introduction section is very poor. Please rewrite this within 1.5 to 2 page giving all the relevant information (Novelty, limitation, application) and recent studies. We believe our introduction (although it does not necessarily follow the structure suggested by the reviewer) clearly covers the context of the work: (1) we identify the types of surface water maps required to help understand and predict disease risk modelling; (2) we summarise past and current RS efforts to deliver this type of data; (3) we highlight the challenges with optical RS data and (4) opportunities with radar data (pointing at examples of work using optical and radar derived surface water information for disease related applications). 

However, as suggested by reviewer 1, we have changed small parts of the introduction to make sure the purpose of the presented work is clearer: The main scope of the paper was to quantify the impact of cloud cover on optically derived water surface maps in a monsoon region by comparing a radar derived water surface map with a popular optically derived water surface product, and to discuss the implications for disease modelling). 

4. Why the study area is selected? put your statement in study area section. We have added the following line: “To contrast between different seasonal surface water dynamics and land cover/use landscape patterns,…” Lines133-134

5. Line 188, (< ~30o) degree? It is now 30o

6. Figure 4b where is legend? The legend is the same for all maps shown in Figure4 and is shown in Figure 4c

7. All of the figure placed in low regulation and unclear. Make sure their better regulation and readable. We have submitted figures with high resolution.

8. Discussion should be rewrite with concise information found from the result. As suggested, we have explicitly pointed to the results where relevant and further improved our discussion.

9. Rewrite the conclusion in a informative way. We have rewritten the conclusions.

10. The title suggests a focus on health issues; therefore, there is a need for greater emphasis on how the work contributes to addressing health concerns. We have changed the title to ensure it better represents the scope of the paper: “Radar versus optical: the impact of cloud when mapping seasonal surface water for health applications in monsoon-affected India.”

11. Add some recent citation This comment is vague; we are not sure which recent citation we should include. However, we have added Chen, S.; Huang, W.; Chen, Y.; Feng, M. An Adaptive Thresholding Approach toward Rapid Flood Coverage Extraction from Sentinel-1 SAR Imagery. Remote Sens. 2021, 13, 4899. https://doi.org/10.3390/rs13234899 as suggested by reviewer 1.

12. Enhance the quality of the writing throughout. We have checked the manuscript for long sentences and grammatical errors and edited as required (see track changes).

---

## [Decision Letter · Decision Letter 1]

23 Sep 2024

PONE-D-23-35272R1Radar versus optical: the impact of cloud when mapping seasonal surface water for health applications in monsoon-affected India.PLOS ONE

Dear Dr. France,

Thank you for submitting your manuscript to PLOS ONE. After careful consideration, we feel that it has merit but does not fully meet PLOS ONE’s publication criteria as it currently stands. Therefore, we invite you to submit a revised version of the manuscript that addresses the points raised during the review process.

We look forward to receiving your revised manuscript.

Kind regards,

Mohammed Sarfaraz Gani Adnan, PhD

Academic Editor

PLOS ONE

Journal Requirements:

Reviewers' comments:

Reviewer's Responses to Questions

**Comments to the Author**

1. If the authors have adequately addressed your comments raised in a previous round of review and you feel that this manuscript is now acceptable for publication, you may indicate that here to bypass the “Comments to the Author” section, enter your conflict of interest statement in the “Confidential to Editor” section, and submit your "Accept" recommendation.

Reviewer #3: (No Response)

Reviewer #4: All comments have been addressed

Reviewer #5: All comments have been addressed

2. Is the manuscript technically sound, and do the data support the conclusions?

Reviewer #3: Yes

Reviewer #4: Yes

Reviewer #5: Yes

3. Has the statistical analysis been performed appropriately and rigorously? 

Reviewer #3: Yes

Reviewer #4: Yes

Reviewer #5: I Don't Know

4. Have the authors made all data underlying the findings in their manuscript fully available?

Reviewer #3: Yes

Reviewer #4: Yes

Reviewer #5: Yes

5. Is the manuscript presented in an intelligible fashion and written in standard English?

Reviewer #3: Yes

Reviewer #4: Yes

Reviewer #5: Yes

6. Review Comments to the Author

Reviewer #3: First off, I was not a reviewer in the previous round and did not read the previous version of this manuscript. I will only discuss the revised version.

Comments:

Generally I think the paper is well-written both linguistically and with regards to its content. I have some minor comments. If these can be addressed I'd recommend acceptance.

-Title: The impact of cloud -> 'the impact of clouds', or 'the impact of cloud cover'

-General: Is disease spread the primary interest when looking at surface water? Are there other reasons we might be interested in this? I can imagine it's also useful for agriculture, drought monitoring, leak detection....?

-Available approach discussion (from line 211 onward):

It doesn't surprise me that bimodal histogram or otsu thresholding are unsuitable, since these methods will fail if you don't have two clear populations of pixels. It does surprise me a bit that Otsu's threshold would set its threshold at the point where it's shown in figure 4c.

-Line 241: Here you state that your threshold is chosen based on whether or not there are many single pixels mapped as water. Do you think these single pixels are actually noise, or might they represent small bodies of water? Or do waterbodies below a certain size threshold not contribute to disease spread? If so, it might also be useful to use a connected components analysis and filter by size. That aside, I agree that the Bayesian approach is probably the better one.

-The Sentinel approach identifies more small waterbodies. How large or small does a waterbody have to be to contribute to disease spread? If there is literature on this it may be useful to put a reference in the discussion.

-Figure 5: What do you mean here by GCP? This is not an acronym I can find elsewhere in the paper.

-I may have missed it in this 94-page pdf, but I can't find any figure descriptions. Unless it conflicts with the guidelines for the journal I'd like to see a brief description of each figure below the figure.

Reviewer #4: I would suggest the authors to add a paragraph in the conclusion section mentioning the limitations and recommendations for future studies. Also, the authors should more clearly explain the novelty or research gap in the abstract. Other than these two points, I believe the authors have addressed the previous comments and the paper is suitable for publishing.

Reviewer #5: The manuscript has improved significantly after the 1st review. However, I have reservations still on one point raised by the previous reviewer also. The title of the manuscript and the discussion of the health related issues in the manuscript. The manuscript is supposed to provide improvements on the models for the mapping of water bodies in the cloud cover time using the SAR data. The method is somewaht improvement from the previous works. But I feel that the previous works on SAR are also quite accurate in mapping the water bodies so why this new complex method? The comparision with optical data does not make much sense here as everyone know optical data will be useless during the cloud time. This if the authors want to prove that the proposed method is more acurate than the previous method they have to compare it will the existing method using the SAR data. Further the application of the produced water map could be anything if authors what it to be only in the case of health related issues they will have to quantify the same i.e. what will be the quantitative difference when this map is used in standerd health model or if the other water maps are used. This it is recommended to remove this part completely from the manuscript.

7. PLOS authors have the option to publish the peer review history of their article (what does this mean?). If published, this will include your full peer review and any attached files.

Reviewer #3: No

Reviewer #4: No

Reviewer #5: No

---

## [Author Response · Author response to Decision Letter 1]

1 Nov 2024

We thank the reviewers for their valuable feedback and have made further changes following this feedback. 

Reviewer #3

-Title: The impact of cloud -> 'the impact of clouds', or 'the impact of cloud cover'

Title now says 'the impact of cloud cover'

-General: Is disease spread the primary interest when looking at surface water? Are there other reasons we might be interested in this? I can imagine it's also useful for agriculture, drought monitoring, leak detection....?

Reviewer#3 (and reviewer#5) is right to point out that mapping surface water is relevant to other applications. However, because the mapping work presented here was carried out as an integral part of a disease risk project in India, our focus was on quantifying the loss of information in an area where cloud cover is seasonal and highlighting the consequences this may have for this particular application. A comprehensive discussion acknowledging a range of applications would have required us to first establish in detail the type of surface water information required by each application (in terms of repeat frequency and spatial resolution) followed by application specific discussions covering the implications and solutions. An interesting proposition, however, not within the scope of the presented work. However, to highlight our work’s relevance to other applications, we have added the following sentence (highlighted in yellow) at the beginning of our discussion:

Line 417: “Accurate and high temporal and spatial resolution surface water mapping is vital for modelling the risks of wide-ranging water-related diseases.  It is also relevant to other applications requiring seasonal mapping, such as, monitoring water availability in water scares regions, or quantifying the contribution of seasonal and/or small lakes as carbon emitters (66) “ 

-Available approach discussion (from line 211 onward):

It doesn't surprise me that bimodal histogram or otsu thresholding are unsuitable, since these methods will fail if you don't have two clear populations of pixels. It does surprise me a bit that Otsu's threshold would set its threshold at the point where it's shown in figure 4c.

It surprised us as well. The Otsu threshold was implemented independently by two of our team members with the same result, giving us confidence figure 4c is correct. 

-Line 241: Here you state that your threshold is chosen based on whether or not there are many single pixels mapped as water. Do you think these single pixels are actually noise, or might they represent small bodies of water? Or do waterbodies below a certain size threshold not contribute to disease spread? If so, it might also be useful to use a connected components analysis and filter by size. That aside, I agree that the Bayesian approach is probably the better one.

Radar backscatter is inherently noisy and it is standard to filter (average) across space or time to reduce the noise. Here we applied a spatial 3x3 lee filter with limited smoothing reducing some but not all of the spatial noise. So, yes the reviewer is correct to wonder why single pixels were considered as noise. After all it is important to map as small a water body as possible, as every water body is important when modelling disease spread. In practice we are compromising between dealing with a noisy data set (even after filtering) and trying to correctly map as many single pixel water bodies as possible. The Bayesian approach does not exclude single pixels, rather it assigns likelihood based on total water body size histogram valley detection, evidence from reference data collection AND noise detection. So, while small water bodies might look like noise, there are three other ways the approach can raise the likelihood of them being classified as water. In the end what we produced is a water body map that is slightly better at mapping smaller water bodies than a 30m TM based map. 

We have added the following text (highlighted in yellow) - line 439 - further highlighting the importance of high(er) spatial resolution radar compared to high(er) spatial resolution optical:

“The use of higher spatial resolution optical imagery (e.g., from Sentinel-2 or Planet Labs) could resolve this, however, where disease risks are linked to the rainy season, and transmission is linked to small waterbodies, the benefits of using SAR over optical for surface water mapping are particularly pronounced,…”

-The Sentinel approach identifies more small waterbodies. How large or small does a waterbody have to be to contribute to disease spread? If there is literature on this it may be useful to put a reference in the discussion.

According to our social scientists conducting community household surveys in the landscape, people, livestock and wildlife, all of which are hosts for pathogens and vectors, use water-bodies as small or smaller than 10m2 for watering as well as larger water-body edges. So a range of water-body sizes and types will be relevant to environmental transmission, though these relationships are not often empirically described. We have included this in our discussion as follows:

Line 435: “According to our social scientists conducting community household surveys in the landscape, people, livestock and wildlife, all of which are hosts for pathogens and vectors, use waterbodies as small or smaller than 10m2 as well as larger water-body edges for watering. So, capturing the complete range of water-body sizes is key to understand environmental transmission.”

We have also enhanced the importance of access to high spatial resolution by 

1. adding ‘high spatial resolution’ to the following sentence – line 439: “The use of higher spatial resolution optical imagery (e.g., from Sentinel-2 or Planet Labs) could resolve this, however where disease risks are linked to the rainy season, and transmission is linked to small waterbodies, the benefits of using high spatial resolution SAR over optical for surface water mapping are particularly pronounced…”

2. and including in our conclusions – line 521: “Still, without meter resolution SAR imaging, many very small water bodies, relevant for disease risk modelling and other applications, will remain unmapped.” 

-Figure 5: What do you mean here by GCP? This is not an acronym I can find elsewhere in the paper.

Thank you for pointing this out. We have updated the figure, replacing GCP (ground control points) with GRP (ground reference points) and included the acronym in the figure title: 

“ Fig 5. Distribution of ground reference points (GRF) displayed with the S1A water body map…”

-I may have missed it in this 94-page pdf, but I can't find any figure descriptions. Unless it conflicts with the guidelines for the journal I'd like to see a brief description of each figure below the figure.

This is to do with how the material is being submitted online. While the manuscript itself contains the figure titles, the figures are submitted without title.

Reviewer #4: 

I would suggest the authors to add a paragraph in the conclusion section mentioning the limitations and recommendations for future studies.

We have added a final conclusion at the end:

Line 521: “Still, without meter resolution SAR imaging, many very small water bodies, relevant for disease risk modelling and other applications, will remain unmapped.“ 

Also, the authors should more clearly explain the novelty or research gap in the abstract. 

We have edited the abstract to emphasize the lack of studies quantifying the impact of cloud cover: “A critical drawback of the latter is data loss due to cloud cover, however few studies have quantified this.”

Reviewer #5: 

The manuscript has improved significantly after the 1st review. However, I have reservations still on one point raised by the previous reviewer also. The title of the manuscript and the discussion of the health related issues in the manuscript. The manuscript is supposed to provide improvements on the models for the mapping of water bodies in the cloud cover time using the SAR data. The method is somewaht improvement from the previous works. But I feel that the previous works on SAR are also quite accurate in mapping the water bodies so why this new complex method? The comparision with optical data does not make much sense here as everyone know optical data will be useless during the cloud time. This if the authors want to prove that the proposed method is more acurate than the previous method they have to compare it will the existing method using the SAR data. 

Actually, our main focus of the presented paper is quantifying data loss caused by cloud cover in optical imagery. The most important point is that when an application (in our case disease risk mapping) requires seasonal mapping in regions where cloud cover is dense, the consequences of relying on optically derived maps is substantial. As far as we are aware no other published study has quantified the data loss. We felt it was important to highlight this as, although the impact of cloud cover on optical imagery may be obvious to remote sensing experts, this is not necessarily the case for non-experts, and many are using optically derived products in their applications. We have made this message stronger in the abstract and conclusions as follows: 

Abstract: “A critical drawback of the latter is data loss due to cloud cover, however few studies have quantified this.”

Conclusions : “Even though the 30m JRC Global Surface Water product achieves high surface water mapping accuracies when there is no cloud, using Sentinel-1A SAR data becomes essential for areas with highly dynamic surface waters and cloudy/rainy seasons as the impact of cloud cover on optically derived seasonal information is substantial.”

Although the core of the work presented is a RS-based study, the main, initial objective was not to develop a new (better) method using SAR, but to apply an easy to implement SAR based alternative for disease risk modelling and to quantify the loss of information due to cloud. The Bayesian approach evolved to check the validity of going manual and consequently an important main point is that manual is as good as many of the other approaches developed to threshold SAR data. Also, the Bayesian approaches (including ours) require priors which are often manually generated. Finally, we do not accept our Bayesian approach as more complex than any of the others suggested. As a matter of fact, our approach (e.g., applicable to other landscapes in the future) has the potential to remove the need to tile and search for bi-modal distributions, making its implementation less complex.

Further the application of the produced water map could be anything if authors what it to be only in the case of health related issues they will have to quantify the same i.e. what will be the quantitative difference when this map is used in standerd health model or if the other water maps are used. This it is recommended to remove this part completely from the manuscript.

See our reply to Reviewer#3. Also the reviewer’s suggestion to quantify the impact of the data loss on the disease risk models, although an interesting follow-on, would substantially expand the work and is therefore not within the scope of this particular paper.

---

## [Decision Letter · Decision Letter 2]

5 Nov 2024

Radar versus optical: the impact of cloud cover when mapping seasonal surface water for health applications in monsoon-affected India.

PONE-D-23-35272R2

Dear Dr. France,

We’re pleased to inform you that your manuscript has been judged scientifically suitable for publication and will be formally accepted for publication once it meets all outstanding technical requirements.

Kind regards,

Mohammed Sarfaraz Gani Adnan, PhD

Academic Editor

PLOS ONE

Additional Editor Comments (optional):

Reviewers' comments:

Reviewer's Responses to Questions

**Comments to the Author**

1. If the authors have adequately addressed your comments raised in a previous round of review and you feel that this manuscript is now acceptable for publication, you may indicate that here to bypass the “Comments to the Author” section, enter your conflict of interest statement in the “Confidential to Editor” section, and submit your "Accept" recommendation.

Reviewer #3: All comments have been addressed

2. Is the manuscript technically sound, and do the data support the conclusions?

Reviewer #3: Yes

3. Has the statistical analysis been performed appropriately and rigorously? 

Reviewer #3: Yes

4. Have the authors made all data underlying the findings in their manuscript fully available?

Reviewer #3: Yes

5. Is the manuscript presented in an intelligible fashion and written in standard English?

Reviewer #3: Yes

6. Review Comments to the Author

Reviewer #3: All of my comments were addressed. There is one tiny issue in the legend for fig. 5:

Fig 5. Distribution of ground reference points (GFP) displayed

I guess this would have to be 'GRP'?

7. PLOS authors have the option to publish the peer review history of their article (what does this mean?). If published, this will include your full peer review and any attached files.

Reviewer #3: No

---

## [Editor Report · Acceptance letter]

13 Nov 2024

PONE-D-23-35272R2 

PLOS ONE

Dear Dr. Gerard, 

I'm pleased to inform you that your manuscript has been deemed suitable for publication in PLOS ONE. Congratulations! Your manuscript is now being handed over to our production team.

Kind regards, 

on behalf of

Dr. Mohammed Sarfaraz Gani Adnan 

Academic Editor

PLOS ONE